# Protein phosphatase 1 regulates atypical mitotic and meiotic division in *Plasmodium* sexual stages

Mohammad Zeeshan[1], Rajan Pandey[1,11], Amit Kumar Subudhi [2,11], David J. P. Ferguson [3,4,11], Gursimran Kaur[1], Ravish Rashpa[5], Raushan Nugmanova[2], Declan Brady[1], Andrew R. Bottrill [6], Sue Vaughan[4], Mathieu Brochet [5], Mathieu Bollen [7], Arnab Pain [2,8], Anthony A. Holder [9], David S. Guttery[1,10] & Rita Tewari [1✉]

PP1 is a conserved eukaryotic serine/threonine phosphatase that regulates many aspects of mitosis and meiosis, often working in concert with other phosphatases, such as CDC14 and CDC25. The proliferative stages of the malaria parasite life cycle include sexual development within the mosquito vector, with male gamete formation characterized by an atypical rapid mitosis, consisting of three rounds of DNA synthesis, successive spindle formation with clustered kinetochores, and a meiotic stage during zygote to ookinete development following fertilization. It is unclear how PP1 is involved in these unusual processes. Using real-time live-cell and ultrastructural imaging, conditional gene knockdown, RNA-seq and proteomic approaches, we show that *Plasmodium* PP1 is implicated in both mitotic exit and, potentially, establishing cell polarity during zygote development in the mosquito midgut, suggesting that small molecule inhibitors of PP1 should be explored for blocking parasite transmission.

[1] School of Life Sciences, University of Nottingham, Nottingham, UK. [2] Pathogen Genomics Group, BESE Division, King Abdullah University of Science and Technology (KAUST), Thuwal, Kingdom of Saudi Arabia. [3] Nuffield Department of Clinical Laboratory Sciences, University of Oxford, John Radcliffe Hospital, Oxford, UK. [4] Department of Biological and Medical Sciences, Faculty of Health and Life Science, Oxford Brookes University, Gipsy Lane, Oxford, UK. [5] Department of Microbiology and Molecular Medicine, Faculty of Medicine, University of Geneva, Geneva, Switzerland. [6] School of Life Sciences, Gibbet Hill Campus, University of Warwick, Coventry, UK. [7] Laboratory of Biosignaling and Therapeutics, KU Leuven Department of Cellular and Molecular Medicine, University of Leuven, Leuven, Belgium. [8] Research Center for Zoonosis Control, Global Institution for Collaborative Research and Education (GI-CoRE); Hokkaido University, Sapporo, Japan. [9] Malaria Parasitology Laboratory, The Francis Crick Institute, London, UK. [10] Leicester Cancer Research Centre, University of Leicester, Leicester, UK. [11] These authors contributed equally: Rajan Pandey, Amit Kumar Subudhi, David J. P. Ferguson. ✉email: rita. tewari@nottingham.ac.uk

Cell cycle progression involves sequential and highly ordered DNA replication and chromosome segregation in eukaryotes[1,2], which is tightly controlled and coordinated by reversible protein phosphorylation catalysed by protein kinases (PKs) and protein phosphatases (PPs)[3]. Numerous studies have highlighted the importance of the phosphoprotein phosphatase (PPP) family in regulating mitosis in model eukaryotic organisms, often working in conjunction with CDC25 and CDC14 phosphatases, as key regulators of mitotic entry and exit, respectively[4–7].

Protein Phosphatase 1 (PP1) is a member of the PPP family and is expressed in all eukaryotic cells. It plays a key role in the progression of mitosis through dephosphorylation of a large variety of proteins, including mitotic kinases (such as cyclin-dependent kinase 1 (CDK1)[8–11], and regulators of chromosome segregation[12]. Chromosome segregation is orchestrated at the kinetochore, a protein complex that assembles on the centromeres, located at the constriction point of sister chromatids to facilitate and monitor attachment of the sister chromatids to spindle microtubules[13,14]. Correct attachment of the kinetochore to spindle microtubules is regulated by the KMN (KNL1, MIS12 and NDC80) protein network[15], which in mammalian cells integrates the activities of at least five protein kinases (including MPS1, Aurora B, BUB1, PLK1 and CDK1) and two protein phosphatases (PP1 and PP2A-B56). The KMN network mediates kinetochore-microtubule attachments and scaffolds the spindle assembly checkpoint (SAC) to prevent chromosome segregation until all sister chromatids are properly connected to the spindle[16]. The orchestration of reversible protein phosphorylation is crucial to control the spatial-temporal progression of the cell cycle and PP1 has a key role in this process, in particular during mitotic exit. Throughout mitosis, PP1 is inhibited by the cyclin-dependent kinase (CDK1)/cyclin B complex and Inhibitor 1[17,18]; however, during mitotic exit concomitant destruction of cyclin B and reduced activity of CDK1 through dephosphorylation by CDC14 results in subsequent reactivation of PP1 via autophosphorylation and completion of mitotic exit[10,17,19–21].

In *Plasmodium*, the causative organism of malaria, there is a single PP1 orthologue, which is expressed throughout the parasite's complex life-cycle[22] and located in both nucleus and cytoplasm[23]. *P. falciparum* PP1 (*Pf*PP1) shares 80% identity with human PP1c, and likely has a conserved tertiary and secondary structure containing 9 α-helices and 11 β-strands[24]. However, *Pf*PP1 lacks part of the 18 amino-acid motif at the C-terminus of human PP1, which contains a threonine residue (Thr320) that is phosphorylated by CDC2 kinase[25] and dynamically regulates entry into mitosis[24]. Previous studies determined that *P. falciparum* PP1 functionally complements the *Saccharomyces cerevisiae* glc7 (PP1) homologue[26], and subsequent phylogenetic analyses revealed that homologues of the phosphatases CDC14 and CDC25 are absent from *Plasmodium*[23]. Genetic screens and inhibitor studies have shown that PP1 is essential for asexual blood stage development[23,27–29] (also reviewed[24]), in particular for parasite egress from the host erythrocyte[30]. Most studies in *Plasmodium* have been focused on these asexual blood stages of parasite proliferation, and very little is known about the importance of PP1 for transmission stages within the mosquito vector. Recent phosphoproteomic studies and chemical genetics analysis have identified a number of potential *Plasmodium falciparum* (Pf) PP1 substrates modified during egress, including a HECT E3 protein-ubiquitin ligase and GCα, a guanylyl cyclase with a phospholipid transporter domain[30]. Other biochemical studies in *Plasmodium* have identified numerous PP1-interacting partners (PIPs) that are structurally conserved and regulate PP1 activity, including LRR1 (a human SDS22 orthologue), Inhibitor 2 and Inhibitor 3. Other PIPs including eif2β, and GEXP15 have also

been identified[9,24,31–33]. Synthetic peptides containing the RVxF motif of Inhibitor 2 and Inhibitor 3, and the LRR and the LRR cap motif of *Pf*LRR1 significantly reduce *P. falciparum* growth in vitro[34,35] and regulate *Pf*PP1 phosphatase activity[35–37]. However, little is known regarding how PP1 is involved in regulating mitosis and meiosis in the absence of CDC14 and CDC25.

The *Plasmodium* life-cycle is characterised by several mitotic stages and a single meiotic stage (reviewed in refs. [38–40]). However, in this organism kinetochore dynamics and chromosome segregation are poorly understood, especially in mitosis during male gametogenesis (gametogony). This is a very rapid process with three rounds of spindle formation and genome replication from 1N to 8N within 12 min. In addition, little is known about the first stage of meiosis that occurs during the zygote to ookinete transition. We have followed recently the spatio-temporal dynamics of NDC80 throughout mitosis in schizogony, sporogony and male gametogenesis (gametogony), and during meiosis in ookinete development[41], using approaches that offer the opportunity to study the key molecular players in these crucial stages of the life cycle. Although PP1 has been partially characterised and shown to have an essential role during asexual blood stage development in the vertebrate host[23], the role and importance of PP1 during sexual stage development in the mosquito is completely unknown.

Here, using the *Plasmodium berghei* (Pb) mouse model of malaria, we determined the importance of PP1 during the sexual stages within the mosquito vector. PP1 expression and location were studied using the endogenous, GFP-tagged protein and co-localisation with the kinetochore marker, NDC80, to follow progression through chromosome segregation during male gamete formation and zygote differentiation. Using a conditional gene knockdown approach we examined how PP1 orchestrates atypical mitosis and meiosis, and investigated the ultrastructural consequences of PP1 gene knockdown for cell morphology, nuclear pole multiplication and flagella formation during male gamete formation. RNA-Seq analysis was used to determine the consequence of PP1 gene knockdown on global transcription, which disclosed a marked differential expression of genes involved in reversible phosphorylation, motor activity and the regulation of cell polarity. Proteomics studies identified motor protein kinesins as interacting partners of PP1 in the gametocyte. The knockdown of PP1 gene expression blocks parasite transmission by the mosquito, showing that this protein has a crucial function in *Plasmodium* sexual development during both mitosis in male gamete formation and meiosis during zygote to ookinete differentiation.

## Results

**PP1-GFP has a diffuse subcellular location during asexual blood stage schizogony and forms discrete foci during endomitosis.** To examine the spatio-temporal expression of *Plasmodium* PP1 in real-time during cell division, we generated a transgenic *P. berghei* line expressing endogenous PP1 with a C-terminal GFP tag. An in-frame *gfp* coding sequence was inserted at the 3′ end of the endogenous *pp1* locus using single crossover homologous recombination (Fig S1a), and successful insertion was confirmed by diagnostic PCR (Fig S1b). Western blot analysis of schizont protein extracts using an anti-GFP antibody revealed a major 62-kDa band, the expected size for PP1-GFP protein, compared to the 29 kDa WT-GFP (Fig S1c).

During asexual blood stage proliferation, schizogony is characterised by multiple rounds of asynchronous nuclear division without chromosome condensation or cytokinesis. Nuclear division is a closed mitotic process without dissolution-reformation of the nuclear envelope and with the spindle-pole

body (SPB)/microtubule-organising centre (MTOC) embedded within the nuclear membrane[42]. It results in a multinucleated coenocyte termed a schizont, which is resolved by cytokinesis at the end of schizogony into individual merozoites. Live-cell imaging of *P. berghei* asexual blood stages revealed a diffuse cytoplasmic and nuclear distribution of PP1-GFP, together with a distinct single focus of concentrated PP1-GFP in the nucleus of early trophozoite stage (Fig. 1a), representing the early S-phase of the cell cycle when DNA synthesis starts. As schizogony proceeds, the diffuse distribution of PP1-GFP remained; however, in early schizonts each cell displayed two distinct PP1-GFP foci in close association with the stained nuclear DNA. These pairs of PP1-GFP foci became clearer during the middle and late schizont stages but following merozoite maturation and egress the intensity of the foci diminished (Fig. 1a).

To study further the location of the PP1-GFP foci throughout mitosis, we generated by genetic cross parasite lines expressing either PP1-GFP and the kinetochore marker NDC80-mCherry[41] or PP1-GFP and the inner membrane complex (IMC)-associated myosin A (MyoA)-mCherry. Live-cell imaging of these lines revealed co-localisation of PP1-GFP and NDC80-mCherry foci close to the DNA of the nucleus through blood stage development, and especially during late schizogony and segmentation (Fig. 1b); whereas the PP1-GFP foci at the outer periphery of the cell showed a partial co-location with MyoA-mCherry, and only during mid- to late schizogony (Fig. 1c). Thus, in schizonts there is a concentration of PP1-GFP at the kinetochore and at a transient peripheral location during specific stages of schizogony, as well as the diffuse distribution throughout the cytoplasm (Fig. 1).

## PP1-GFP is enriched on kinetochores during chromosome segregation associated with putative mitotic exit in male gametogony.

To study PP1 expression and location through the three rounds of DNA replication and chromosome segregation prior to nuclear division during male gametogony, we examined PP1-GFP by live-cell imaging over a 15-min period following male gametocyte activation. Following activation with xanthurenic acid and decreased temperature, male gametocytes undergo three rounds of DNA replication and mitotic division followed by chromosome condensation and exflagellation, resulting in eight gametes[43]. Before activation (0 min), PP1-GFP was detected with a diffuse location throughout gametocytes (Fig. 2a). At one minute post-activation, PP1-GFP accumulated at one end of the nucleus as a single focal point (Fig. 2a). After 2–3 min two distinct foci were observed on one side of the nucleus, concurrent with the first round of chromosome replication/segregation (Fig. 2a). Subsequently, four and eight PP1-GFP foci were observed at 6–8 min and 8–12 min post-activation, respectively, corresponding to the second and third rounds of chromosome replication/segregation. These discrete PP1-GFP foci dispersed prior to karyokinesis and exflagellation of the mature male gamete 15 min post-activation, leaving residual protein remaining diffusely distributed throughout the remnant gametocyte and flagellum (Fig. 2a).

To investigate further the location of PP1-GFP during spindle formation and chromosome segregation, the parasite line expressing both PP1-GFP and NDC80-mCherry was examined by live-cell imaging to establish the spatio-temporal relationship of the two proteins. We found that the discrete PP1-GFP foci colocalized with NDC80-mCherry at different stages of male gametogony, including when up to eight kinetochores were visible (Fig. 2b), but PP1-GFP was not associated with the arc-like bridges of NDC80-mCherry representing the spindle[41] at 2 min and 4–5 min post activation, suggesting that PP1-GFP is only

increased at the kinetochore during initiation and termination of spindle division (Fig. 2b).

## PP1-GFP may determine apical polarity during zygote-ookinete development and has a nuclear location on kinetochores during meiosis.

After fertilisation in the mosquito midgut, the diploid zygote differentiates into an ookinete within which the first stage of meiosis occurs. During this process DNA is duplicated to produce a tetraploid cell with four haploid genomes within a single nucleus in the mature ookinete. PP1 is known to be crucial during meiotic chromosome segregation[12], and therefore we analysed the spatio-temporal expression of PP1-GFP during ookinete development using live-cell imaging. PP1-GFP was expressed in both male and female gametes with a diffuse distribution, along with a single focus of intense fluorescence at one end (potentially the basal body) of each male gamete (Fig. 3a). Initially, the zygote also had a diffuse PP1-GFP distribution, but after 2 h (stage I) an enriched focus developed at the periphery of the zygote, marking the point that subsequently protruded out from the cell body and developed into the apical end of the ookinete. A strong PP1-GFP fluorescence signal remained at this apical end throughout ookinete differentiation (Fig. 3a). In addition, PP1-GFP was enriched in the nucleus at four discrete foci in mature ookinetes (Fig. 3a). Analysis of the parasite line expressing both PP1-GFP and NDC80-mCherry showed that these four foci correspond to the kinetochores of meiotic development in the ookinete (Fig. 3b).

## PP1-GFP has a diffuse distribution with multiple nuclear foci during oocyst development, representing endomitosis.

Upon maturation the ookinete penetrates the mosquito midgut wall and embeds in the basal lamina to form an oocyst. Over the course of 21-days sporogony occurs in which several rounds of closed endomitotic division produce a multiploid cell termed a sporoblast[44]; with subsequent nuclear division resulting in thousands of haploid nuclei and concomitant sporozoite formation. We observed multiple foci of PP1GFP along with diffused localisation during oocyst development (Fig. 3c). Dual colour imaging showed a partial co-localisation of nuclear PP1-GFP and NDC80-mCherry foci throughout oocyst development and in sporozoites (Fig. 3d), suggesting that PP1-GFP is diffusely distributed during oocyst development but also recruited to kinetochores, as highlighted by multiple nuclear foci during oocyst development.

## Generation of transgenic parasites with conditional knockdown of PP1.

Our previous systematic analysis of the *P. berghei* protein phosphatases suggested that PP1 is likely essential for erythrocytic development[23], a result which was further substantiated in a recent study of *P. falciparum*[30]. Since little is known about the role of PP1 in cell division during the sexual stages of parasite development in the mosquito vector, we attempted to generate transgenic, conditional knockdown lines using either the auxin inducible degron (AID) or the promoter trap double homologous recombination (PTD) systems. Despite successful generation of a parasite line expressing PP1-AID, addition of indole-3-acetic acid (IAA) did not result in PP1 depletion (Fig S2c) and male gametogony was not affected (Fig S2c). Therefore, we used a promoter trap method that had been used previously for functional analysis of the condensin core subunits SMC2/4 and the APC/C complex component, APC3[45,46]. Double homologous recombination was used to insert the *ama* 1 promoter upstream of *pp1* in a parasite line that constitutively expresses GFP[47]. The expected outcome was a parasite expressing PP1 during asexual blood stage development,

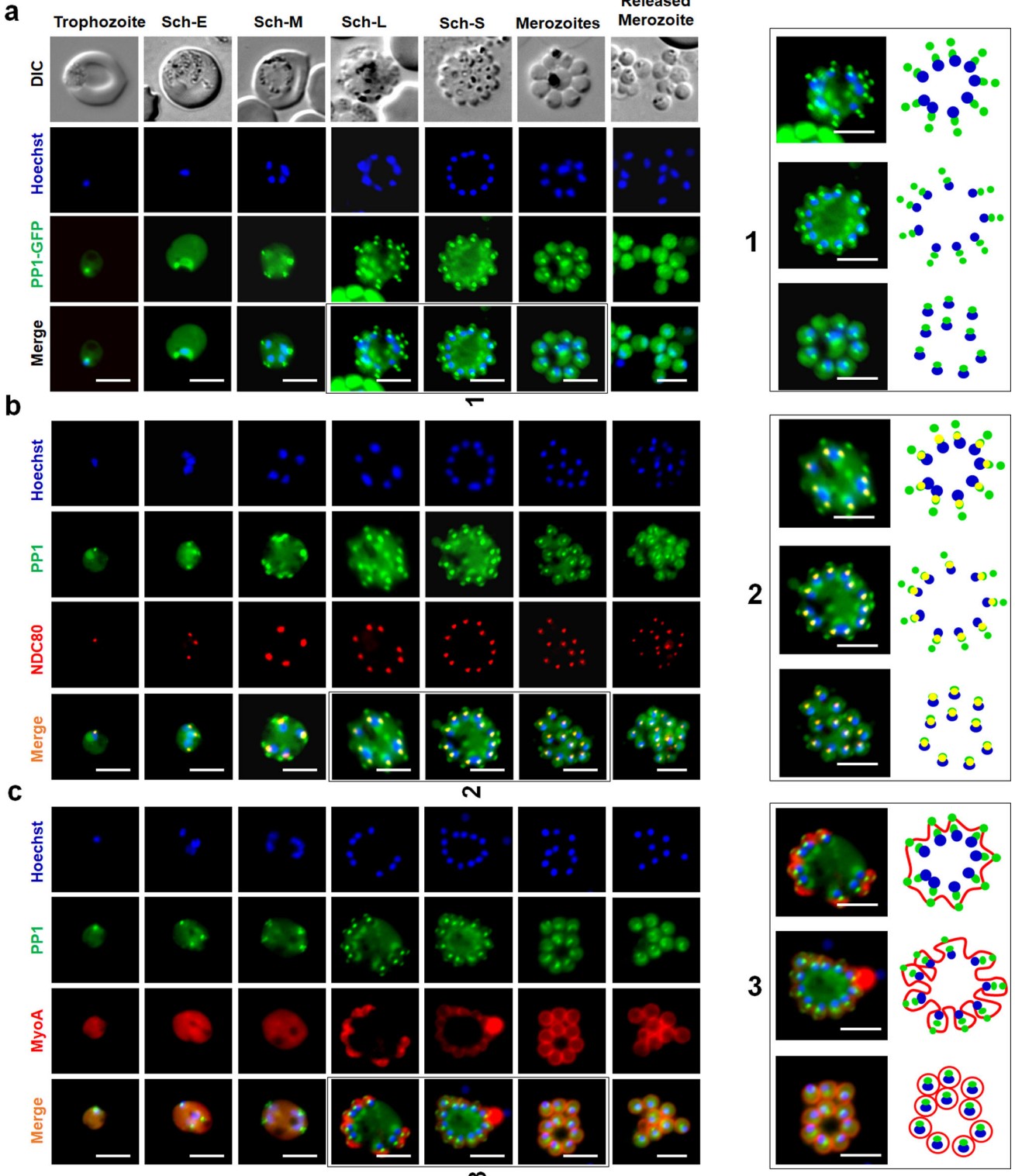

**Fig. 1 Location of PP1 during asexual blood stage schizogony and its association with kinetochore (NDC80) and glideosome (MyosinA). a** Live-cell imaging of PP1-GFP (Green) showing its location during different stages of intraerythrocytic development and in free merozoites. DIC differential interference contrast, Hoechst stained DNA (blue), Merge green and blue images merged. A schematic guide showing the locations of PP1GFP foci during segmentation of merozoites is depicted in right-hand panel. **b** Live-cell imaging showing location of PP1–GFP (green) in relation to the kinetochore marker NDC80-mCherry (red) and DNA (Hoechst, blue). Merge: green, red and blue images merged. A schematic guide showing the locations of PP1GFP foci with NDC80-mCherry during segmentation of merozoites is depicted in right-hand panel. **c** Live imaging showing the location of PP1–GFP (green) in relation to inner membrane complex marker MyoA-mCherry (red) and DNA (Hoechst, blue) during different stages of intraerythrocytic development and in extracellular merozoites. A schematic guide showing the locations of PP1GFP foci with MyoA-mCherry during segmentation of merozoites is depicted in the three right-hand panels. Merge: green, red and blue images merged. Sch-E (Early schizont), Sch-M (Middle schizont), Sch-L (Late schizont) and Sch-S (Segmented schizont). In all panels, scale bar = 5 μm.

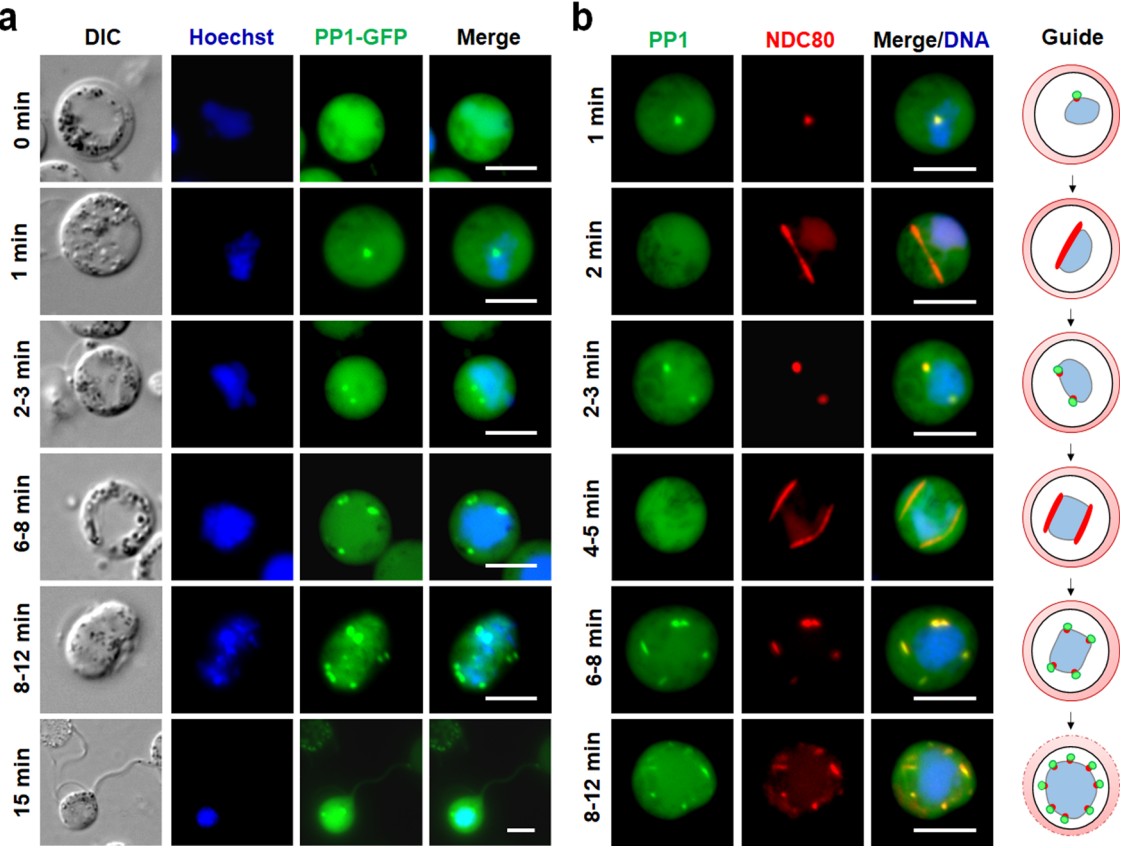

**Fig. 2 The location of PP1 and its association with the kinetochore during chromosome segregation in male gametogony. a** Live-cell imaging of PP1-GFP during male gametogony showing an initial diffused localisation before activation and focal points after activation in the later stages (shown as minutes post activation). Panels are DIC, Hoechst (blue, DNA), PP1-GFP (green) and Merge (green and blue channels). **b** Live-cell imaging of parasite line expressing both PP1-GFP and NDC80-mCherry showing location of PP1 (green) and NDC80 (red) in male gametocytes at different time points after activation. A schematic guide showing the locations of PP1GFP foci with DNA and NDC80-mCherry during male gametogony is depicted in the right panel. Merge/DNA is green, red and blue (Hoechst, DNA) channels. In both panels, scale bar = 5 μm.

but not at high levels during the sexual stages (Fig S2d). This strategy resulted in the successful generation of two independent PP1PTD parasite clones (clones 3 and 5) produced in independent transfections, and integration confirmed by diagnostic PCR (Fig S2e). The PP1PTD clones had the same phenotype and data presented here are combined results from both clones. Quantitative real-time PCR (qRT-PCR) confirmed a downregulation of *pp1* mRNA in gametocytes by ~90% (Fig. 4a).

**Knockdown of PP1 gene expression during the sexual stages blocks parasite transmission by affecting parasite growth and development within the mosquito vector.** The PP1PTD parasites grew slower than the WT-GFP controls in the asexual blood stage (Fig S2f) but produced normal numbers of gametocytes in mice injected with phenylhydrazine before infection. Gametocyte activation with xanthurenic acid in ookinete medium resulted in significantly fewer exflagellating PP1PTD parasites per field in comparison to WT-GFP parasites, suggesting that male gamete formation was severely retarded (Fig. 4b). For those few PP1PTD gametes that were released and viable, zygote-ookinete differentiation following fertilisation was severely affected (Fig. 4c), with significantly reduced numbers of fully formed, banana-shaped ookinetes. In all PP1PTD samples analysed 24 h post-fertilisation, the vast majority of zygotes were still round, and there was a significant number of abnormal retort-shaped cells with a long thin protrusion attached to the main cell body (Fig. 4c, d).

To investigate further the function of PP1 during parasite development in the mosquito, *Anopheles stephensi* mosquitoes were fed on mice infected with PP1PTD and WT-GFP parasites as a control. The number of GFP-positive oocysts on the mosquito gut wall was counted on days 7, 14 and 21 post-infection. No oocysts were detected from PP1PTD parasites; whereas WT-GFP lines produced normal oocysts (Fig. 4e, f), all of which had undergone sporogony (Fig. 4g). Furthermore, no sporozoites were observed in the salivary glands of PP1PTD parasite-infected mosquitoes, in contrast to WTGFP parasite-infected mosquitoes (Fig. 4h). This lack of viable sporozoites was confirmed by bite back experiments that showed no further transmission of PP1PTD parasites, in contrast to the WTGFP lines, which showed positive blood stage infection 4 days after mosquito feed (Fig S2g). All the raw data files can be found in Supplementary data 1.

**Ultrastructure analysis of PP1PTD gametocytes shows defects in nuclear pole and axoneme assembly.** To determine the ultrastructural consequences of reduced PP1 expression, WTGFP and PP1PTD gametocytes were examined at 6- and 30-min post activation (pa) by electron microscopy. At 6 min pa, the WTGFP male gametocytes exhibited an open nucleus with a number of nuclear poles (Fig. 5a). Basal bodies and normal 9 + 2 axonemes were often associated with the nuclear poles (Fig. 5b, c). At 6 min pa, the PP1PTD line exhibited a similar morphology (Fig. 5d–f). However, quantitative analysis showed that PP1PTD gametocytes were relatively less well-developed, with fewer nuclear poles (0.86

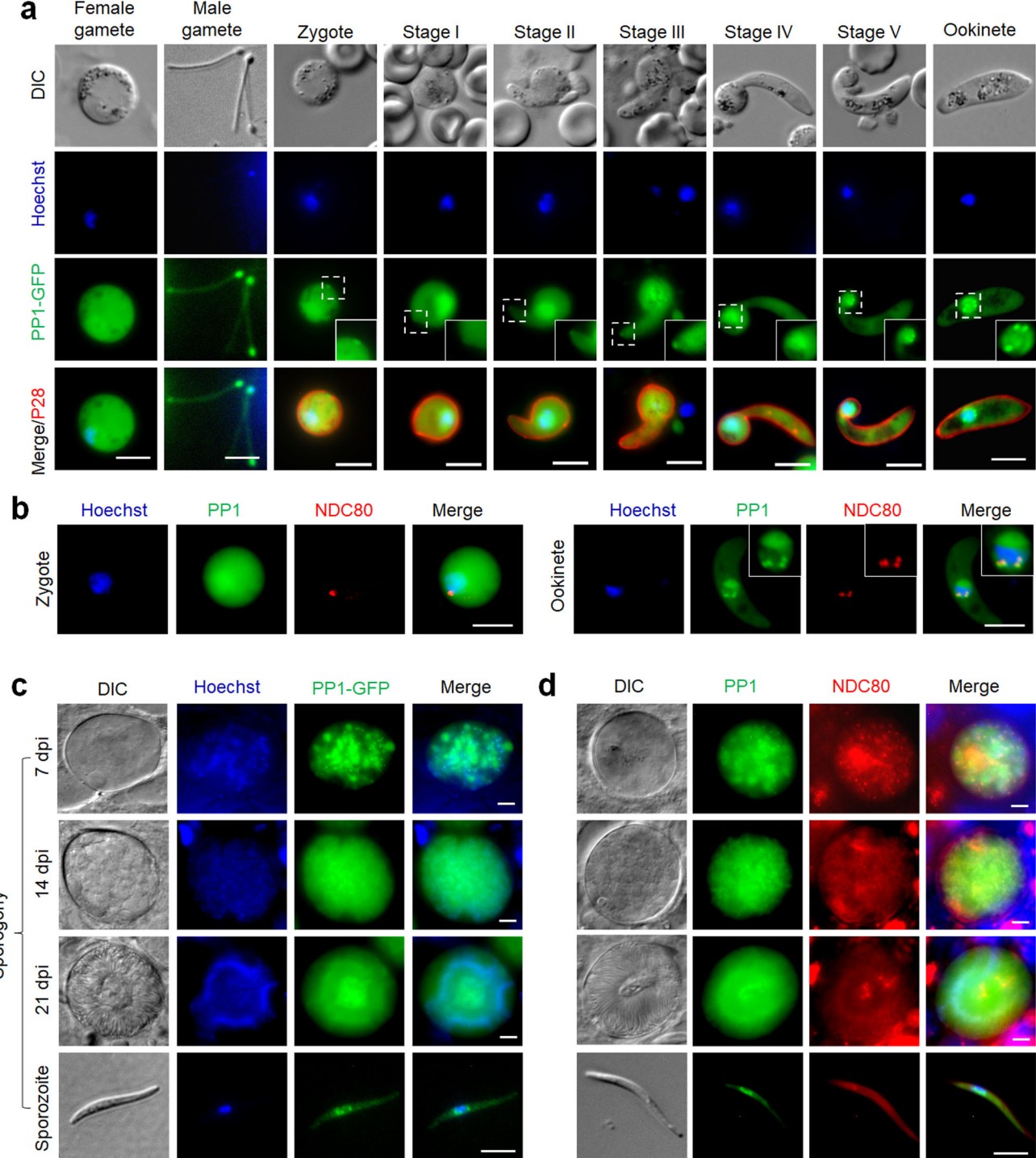

**Fig. 3 PP1-GFP localisation during zygote formation, ookinete development and sporogony inside the mosquito gut. a** Live-cell imaging showing PP1-GFP location in male and female gametes, zygote and during ookinete development (stages I–V and mature ookinete). A cy3-conjugated antibody, 13.1, which recognises the P28 protein on the surface of zygotes and ookinetes, was used to mark these stages. Panels: DIC (differential interference contrast), PP1-GFP (green, GFP), 13.1 (red), Merged: Hoechst (blue, DNA), PP1-GFP (green, GFP) and P28 (red). Scale bar = 5 μm. Insets show, at higher magnification, the PP1-GFP signal on the zygote and developing apical end of early stage 'retorts', and in the nucleus of late retorts and ookinete stage. **b** Live-cell imaging of PP1-GFP (green) in relation to NDC80-mCherry (red) and Hoechst staining (blue, DNA) in zygote and ookinete stages. **c** Live-cell imaging of PP1-GFP in developing oocysts in mosquito guts at 7-, 14- and 21-days post-infection and in a sporozoite. Panels: DIC, Hoechst (blue, DNA), PP1-GFP (green), Merged (blue and green channels). **d** Live-cell imaging of PP1-GFP in relation to NDC80 in developing oocysts and in a sporozoite. Panels: DIC (differential interference contrast), PP1-GFP (green), NDC80-mCherry (red), Merge (Hoechst, blue, DNA; PP1-GFP, green; NDC80-mCherry, red). In all panels, scale bar = 5 μm.

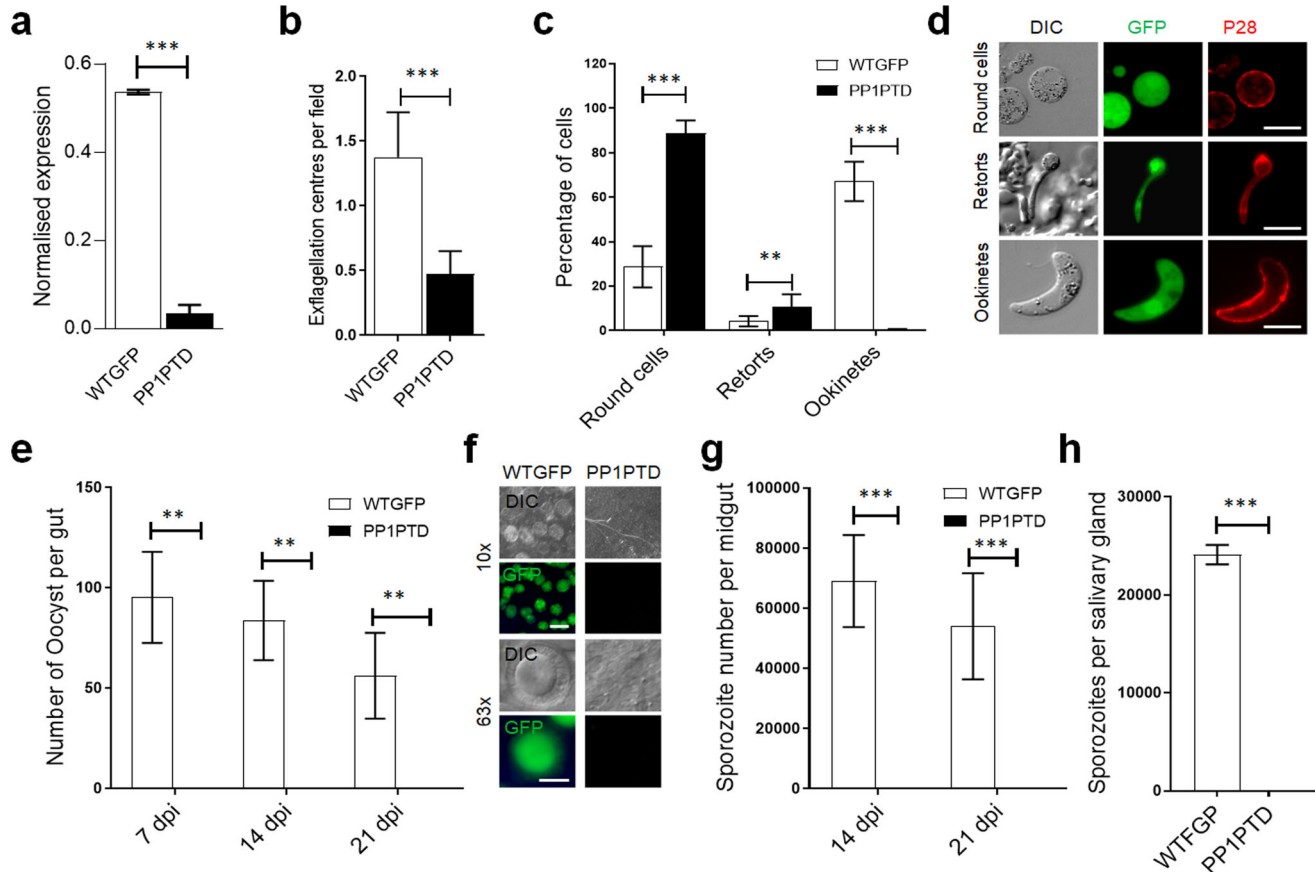

**Fig. 4 PP1 has an important role during male gamete formation and zygote-ookinete development. a** qRT-PCR analysis of *pp1* transcription in PP1PTD and WT-GFP parasites, showing the downregulation of *pp1*. Each bar is the mean of three biological replicates ±SD. **b** Male gametogony (exflagellation) of PP1PTD line (black bar) and WT-GFP line (white bar) measured as the number of exflagellation centres per field. Mean ± SD; *n* = 3 independent experiments. **c** Ookinete conversion as a percentage for PP1PTD and WT-GFP parasites. Ookinetes were identified using P28 antibody as a surface marker and defined as those cells that differentiated successfully into elongated 'banana shaped' ookinetes. Round cells show zygotes that did not start to transform and 'retorts' could not differentiate successfully in ookinetes. Mean ± SD; *n* = 3 independent experiments. **d** Representative images of round cells, retorts and fully differentiated ookinetes. Scale bar = 5 μm. **e** Total number of GFP-positive oocysts per infected mosquito in PP1PTD and WT-GFP parasites at 7-, 14- and 21-days post infection (dpi). Mean ± SD; *n* = 3 independent experiments. **f** Representative images of mosquito midguts on day 14 showing them full of oocysts in WTGFP and no oocyst in PP1PTD. Scale bar = 200 μm for ×10 magnification and 50 μm for ×63 magnification. **g** Total number of sporozoites in oocysts of PP1PTD and WT-GFP parasites at 14 and 21 dpi. Mean ± SD; *n* = 3 independent experiments. **h** Total number of sporozoites in salivary glands of PP1PTD and WT-GFP parasites. Bar diagram shows mean ± SD; *n* = 3 independent experiments. Unpaired *t*-test was performed for statistical analysis. *\*p* < 0.05, *\*\*p* < 0.01, *\*\*\*p* < 0.001.

WTGFP v. 0.56 PP1PTD), basal bodies (1.03 WTGFP v. 0.56 PP1PTD) and axonemes (2.57 WTGFP v. 1.27 PP1PTD), based on random sections of 50 male gametocytes in each group.

At 30 min post activation, most WTGFP male gametocytes (83%) were at a late stage of development with nuclei showing chromatin condensation (Fig. 5g) and examples of exflagellation with the flagellate gamete protruding from the surface (Fig. 5h) and free male gametes (Fig. 5g, i). In contrast, most PP1PTD male gametocytes (85%) were stalled at an early stage of development similar to that at 6 min pa (Fig. 5j–l and Fig S3a), based on random sections of fifty male gametocytes in each group. The main morphological difference between the two parasite lines was a marked increase in the number and length of the axonemes in the PP1PTD parasites at 30 min pa (cf Fig. 5a, j). In summary, the development of PP1PTD male gametocytes was severely retarded, although axoneme growth increased.

**PP1PTD parasites have altered expression of genes involved in cell cycle progression, cell motility and apical cell polarity.** To determine the consequences of PP1 knockdown on global mRNA

transcription, we performed RNA-seq in duplicate on PP1PTD and WT-GFP gametocytes, 0 min and 30 min post-activation (Fig. 6a, b). All replicates were clustered together based on condition (Fig S3b) and totals of 13–32 million RNA-seq reads were generated per sample for the global transcriptome (Fig S3c). In both 0 min and 30 min activated gametocytes, *pp1* was significantly down-regulated (by more than 16-fold, *q* value < 0.05) in PP1PTD parasites compared to WT-GFP parasites (Fig. 6a), thus confirming the qPCR result (Fig. 4a). We observed significant transcriptional perturbation in both 0 min and 30 min activated gametocytes, affecting the expression of 530 and 829 genes respectively (representing ~10% and 16% of the total gene complement) (Supplementary Data 2). Of the total perturbed genes, 344 and 581 were significantly down-regulated and 186 and 248 genes were significantly up-regulated in the PP1PTD parasites compared to WT-GFP controls in 0 min and 30 min activated gametocytes, respectively (Supplementary Data 2).

To explore the biological roles of the genes down-regulated following reduced *pp1* expression, we performed gene-ontology (GO) based enrichment analysis. We observed that many genes encoding kinases, phosphatases and motor proteins were

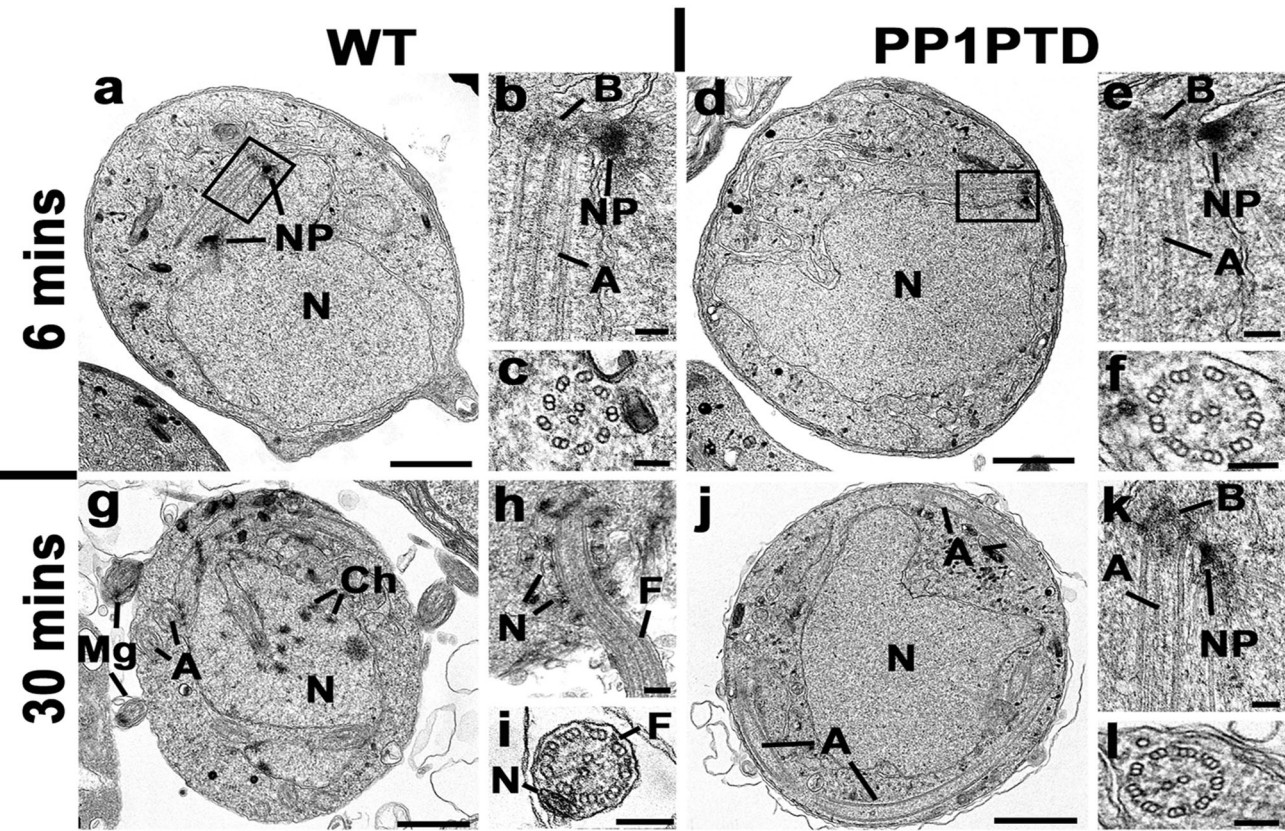

**Fig. 5 Ultrastructure analysis of PP1PTD gametocytes shows defects in nuclear pole and axoneme assembly during male gametogony.** Electron micrographs of WTGFP (**a–c**, **g–i**) and PP1PTD (**d–f**, **j–l**) male gametocytes at 6 min (**a–f**) and 30 min (**g–i**) post activation (pa). Scale bars represent 1 μm (**a**, **d**, **g**, **j**) and 100 nm in all other micrographs. **a** Low power micrograph of a WTGFP male gametocyte with two nuclear poles (NP) associated with the nucleus (N). **b** Enlargement of the enclosed area showing the nuclear pole (NP) with adjacent basal body (B) and associated axoneme (A). **c** Cross section of an axoneme showing the 9 + 2 microtubule arrangement. **d** Low power micrograph of PP1PTD male gametocyte showing the central nucleus (N) with a nuclear pole and associated basal body (enclosed area). **e** Enlargement of the enclosed area showing the nuclear pole (NP) with adjacent basal body (B) and associated axoneme (A). **f** Cross section through an axoneme showing the 9 + 2 arrangement of microtubules. **g** Low power micrograph of a 30-min pa WTGFP male gametocyte showing the nucleus with areas of condensed chromatin. Note the cross sectioned free male gametes (Mg). A – axoneme. **h** Periphery of a male gametocyte undergoing exflagellation with the nucleus (N) associated with flagellum (F) protruding from the surface. **i** Cross section of a free male gamete showing the 9 + 2 microtubules of the flagellum (F) and electron dense nucleus (N). **j** Low power micrograph of a PP1PTD male gametocyte at 30 min pa showing the central nucleus (N) with a nuclear pole and an increased number of axoneme profiles (A) within the cytoplasm. **k** Detail of the periphery of a nucleus showing the nuclear pole (NP), basal body (B) and associated axoneme (A). **l** Cross section of an axoneme showing the 9 + 2 microtubule arrangement.

differentially expressed in either or in both 0 min or 30 mins activated gametocytes (Fig. 6c), complementing the observations from our phenotypic analysis (Fig. 4c, d). In addition, we also observed that many genes encoding proteins involved in cell motility, cell-cycle progression, host-cell entry and ookinete development were significantly affected in activated PP1PTD gametocytes. Transcript levels measured by RNA-seq were further validated by qRT-PCR for a few selected genes modulated in gametocytes at 0 min and 30 min, and involved in regulation of cell cycle, cell motility, and ookinete invasion (Fig. 6d).

**PP1–GFP interacts with similar proteins in schizonts and gametocytes, but with a predominance of microtubule motor kinesins in gametocytes.** Previous studies have analysed the PP1 interactome in *P. falciparum* schizonts, revealing several interacting partners[32,48]. Here, we analysed the PP1 interactome in *P. berghei* schizonts and gametocytes to establish whether there were differences that might reflect distinct functions in the two stages. We immunoprecipitated PP1-GFP from lysates of schizonts following parasite culture in vitro for 10 h and 24 h and from lysates of gametocytes 10–11 min post activation because of high PP1-

GFP abundance at these stages (Fig. 7a and Supplementary Data 3). Mass spectrometric analysis of these pulldowns identified several proteins common to both schizont and gametocyte stages, suggesting similar protein complexes in both stages (Fig. 7a, b). In addition, we also identified in the pulldown from gametocyte lysates a number of microtubule proteins that are associated with the spindle or axoneme, including kinesin-8B, kinesin-15, kinesin-13 and PF16 (Fig. 7a, b). These proteins are specific to male gametocytes and may have important roles in axoneme assembly and male gametogony[43,49].

## Discussion

Reversible protein phosphorylation is crucial for cell cycle progression in eukaryotes and is tightly controlled by a variety of protein kinases and phosphatases[50,51]. *Plasmodium* possesses a set of divergent protein kinases and protein phosphatases, which regulate many processes during cell division and parasite development throughout different stages of the life-cycle[23,52], and PP1 is quantitatively one of the most important protein phosphatases that hydrolyse serine/threonine linked phosphate ester bonds[53]. It is expressed in all cells and is highly conserved in organisms

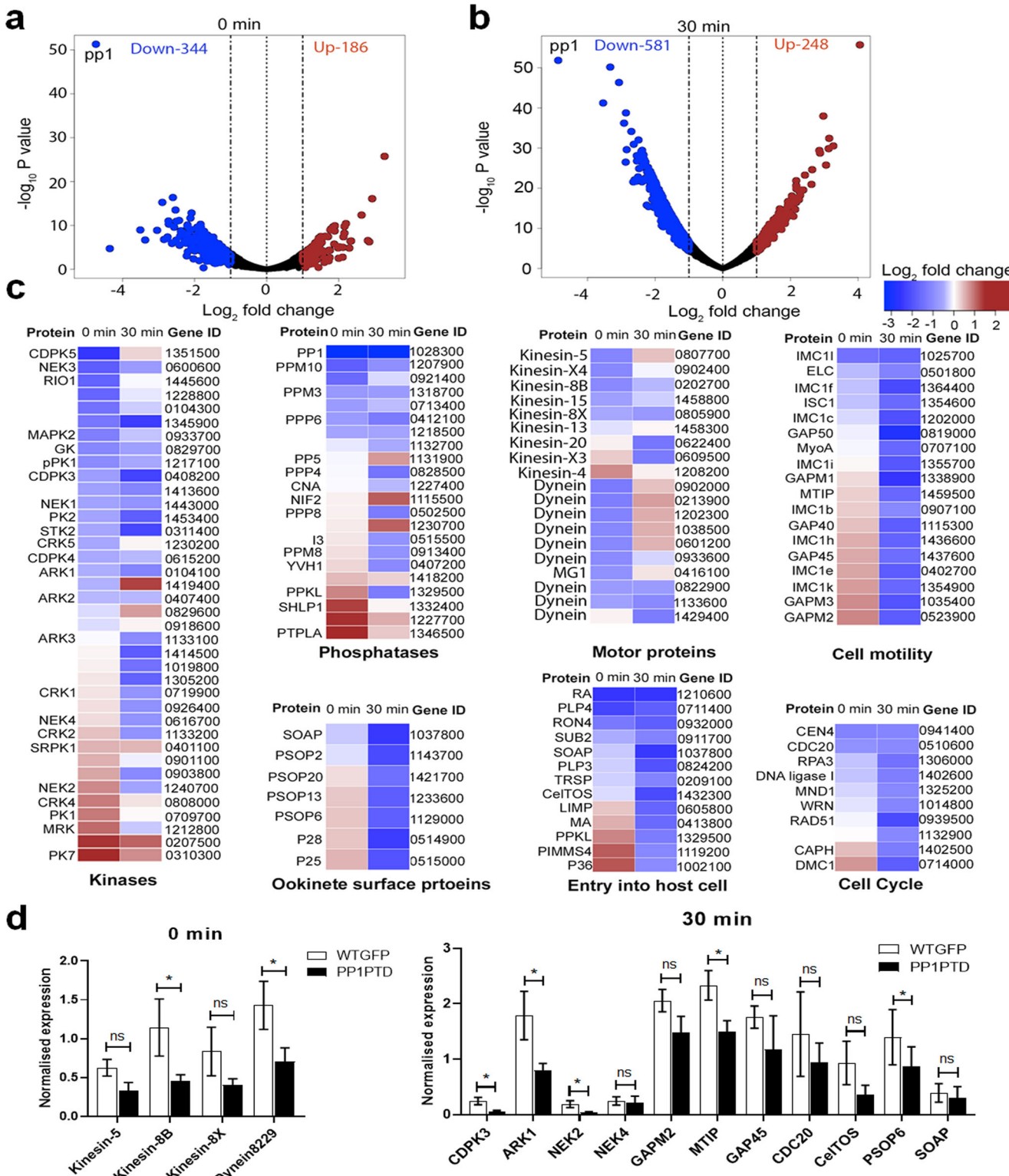

**Fig. 6 Transcriptome of PP1PTD mutant reveals important roles of PP1 in parasite cell cycle, motor protein function and cell polarity during gametocyte biology.** Volcano plots showing significantly down-regulated (blue $Log_2$ fold $\leq -1$, $q$ value < 0.05) and up-regulated genes (brown $Log_2$ fold $\geq 1$, $q$ value < 0.05) in PP1PTD compared to wild-type lines in non (0 min)-activated gametocytes (**a**) and 30 min activated gametocytes (**b**). Non-differentially regulated genes are represented as black dots. **c** Expression heat maps showing affected genes from specific functional classes of proteins such as kinases, phosphatases, motor proteins and proteins associated with parasite motility, entry into the host cell, ookinete surface and cell cycle. Genes are ordered based on their differential expression pattern in non-activated gametocytes. **d** Validation by qRT-PCR of a few genes randomly selected based on the RNA-seq data. All the experiments were performed three times in duplicate with two biological replicates. *$p \leq 05$.

**a**

| P. berghei Gene ID | Product Description | Gene names | MW (kDa) | Schizonts- 24 h PP1-GFP | | | GFP | Schizonts-10 h PP1-GFP | | | GFP | Gametocytes-10 min PP1-GFP | | | GFP |
|---|---|---|---|---|---|---|---|---|---|---|---|---|---|---|---|
| PBANKA_1028300 | serine/threonine protein phosphatase PP1 | PP1 | 34963 | 31 | 26 | 23 | 0 | 37 | 26 | 25 | 0 | 23 | 33 | 26 | 0 |
| PBANKA_1218500 | protein phosphatase inhibitor 2, putative | PPI2 | 16664 | 11 | 7 | 9 | 0 | 9 | 8 | 7 | 0 | 10 | 9 | 9 | 0 |
| PBANKA_0516600 | leucine-rich repeat protein | LRR1 | 36735 | 33 | 26 | 26 | 0 | 22 | 29 | 29 | 0 | 11 | 22 | 25 | 0 |
| PBANKA_0601600 | conserved protein, unknown function | NEMF-like | 201925 | 36 | 17 | 35 | 0 | 21 | 37 | 21 | 0 | 12 | 22 | 24 | 0 |
| PBANKA_0515500 | protein phosphatase inhibitor 3, putative | PPI3 | 13065 | 3 | 3 | 3 | 0 | 0 | 4 | 2 | 0 | 2 | 2 | 3 | 0 |
| PBANKA_0515400 | conserved Plasmodium protein | CD2BP2 | 77377 | 17 | 10 | 18 | 0 | 7 | 22 | 12 | 0 | 3 | 2 | 9 | 0 |
| PBANKA_0310800 | RTR1 domain-containing protein, putative | RTR1-like | 97344 | 15 | 8 | 11 | 0 | 19 | 32 | 19 | 0 | 0 | 0 | 7 | 0 |
| PBANKA_1458800 | kinesin, putative | kinesin-15 | 164999 | 0 | 0 | 0 | 0 | 0 | 0 | 0 | 0 | 2 | 7 | 2 | 0 |
| PBANKA_0202700 | kinesin-8, putative | kinesin-8B | 168815 | 0 | 0 | 0 | 0 | 0 | 0 | 0 | 0 | 2 | 4 | 2 | 0 |
| PBANKA_1458300 | kinesin-13, putative | kinesin-13 | 117181 | 0 | 0 | 0 | 0 | 0 | 0 | 0 | 0 | 0 | 3 | 3 | 0 |
| PBANKA_0917400 | armadillo repeat protein PF16 | PF16 | 57748 | 0 | 0 | 0 | 0 | 0 | 0 | 0 | 0 | 0 | 2 | 2 | 0 |

**b**

**Fig. 7 Interacting partners of PP1 during asexual schizont and sexual gametocyte stages. a** List of proteins interacting with PP1 during schizont and gametocyte stages. **b** Venn diagram showing common interacting partners in schizonts and gametocytes with some additional proteins specific to gametocytes.

throughout the Eukaryota including *Plasmodium*[23,54], although a key region at the C-terminus that is known to regulate mitotic exit is missing from the parasite PP1[24]. It forms stable complexes with several PP1-interacting proteins (PIPs) that are diverse and differ in various organisms, control its location and function, and assist in processes throughout the cell cycle[8,55–57].

The cell cycle in *Plasmodium* differs from that of many other eukaryotes with atypical mitotic and meiotic divisions throughout its life cycle[41,43,58]. It lacks several classical cell cycle regulators including the protein phosphatases CDC14 and CDC25[23], and key protein kinases including polo-like kinases[52,59]. Therefore, the role of PP1, or indeed other PPPs in the absence of these protein phosphatases and kinases may be more crucial during cell division in *Plasmodium*. In the present study, we focused on the location and functional role of PP1 in mitotic division, particularly during male gametogony, and in meiosis during the zygote to ookinete transition. Both processes are essential for the sexual stage of the *Plasmodium* life cycle and are required for transmission by the mosquito vector. Our previous systematic functional analysis of the *Plasmodium* protein phosphatome[23] showed that PP1 has an essential role during asexual blood stage development; this finding was supported by a recent study, which showed its role in merozoite egress from the host erythrocyte[30]. Our findings here reveal that PP1 is also expressed constitutively and co-localises with the kinetochore protein NDC80 during different stages of *Plasmodium* asexual and sexual development, hinting at a role during atypical chromosome segregation. Our conditional PP1 gene knockdown suggests that it plays a crucial role in mitotic division during male gametogony; and may regulate cell polarity in meiosis during zygote to ookinete transformation.

Male gametogony in *Plasmodium* is a rapid process with DNA replication and three mitotic divisions followed by karyokinesis and cytokinesis to form eight flagellated male gametes within 15 min of gametocyte activation[41,58]. The expression and localisation profiles show that although the protein is diffusely distributed throughout the cytoplasm and the nucleus, there is a cyclic enrichment of PP1-GFP at the kinetochore, associating with NDC80-mCherry during all three successive genome replication and closed mitotic divisions without nuclear division[41]. The accumulation of PP1-GFP at the kinetochore at the start of nuclear division and subsequent decrease upon completion is similar to the situation in other eukaryotes where the activity of PP1 increases in G2 phase, reduces during prophase and metaphase, and increases again during anaphase[51]. This behaviour

suggests a role for PP1 in regulating rapid mitotic entry and exit during male gametogony. Further analysis of PP1 gene knockdown showed a significant decrease in male gamete formation (i.e. during male gametogony), and ultrastructural analysis revealed fewer nuclear poles and basal bodies associated with axonemes, with a concomitant absence of chromosome condensation in male gametocytes of PP1PTD parasites, suggesting a role for PP1 during chromosome segregation and gamete formation (i.e. flagella formation). A similar phenotype was observed in our recent study of a divergent *Plasmodium* cdc2-related kinase (CRK5)[60]. CRK5 deletion resulted in fewer nuclear poles, and no chromatin condensation, cytokinesis or flagellum formation, suggesting that there may be a coordinated activity of PP1 and CRK5 in reversible phosphorylation. A recent phosphoregulation study across the short time period of male gametogony showed tightly controlled phosphorylation events mediated by several protein kinases including ARK2, CRK5, and NEK1[61]; however, the reciprocal protein phosphatases that reverse these events are unknown. The expression pattern and cyclic enrichment of PP1 at the kinetochores suggest a key reciprocal role of PP1 in reversing the protein phosphorylation mediated by these kinases. This interpretation is supported by our global transcriptome analysis of the PP1PTD parasite, which showed a modulated expression of several serine/threonine protein kinases, protein phosphatases and motor proteins. The similar pattern of reduced expression of some of these protein phosphatases and kinases suggests they have a coordinated role in reversible phosphorylation. Of note, phosphoregulation of motor proteins in *Plasmodium* has been described previously[61] and it plays an important role in spindle assembly and axoneme formation during male gametogony[41,43].

The proteomics analysis showed that PP1 interacts with a conserved set of proteins in both asexual and sexual stages of development, and with additional proteins in gametocytes, implicating major motor proteins that may be required for spindle and axoneme assembly during male gametogenesis (gametogony). Several of the conserved interacting partners also form complexes with PP1 in other eukaryotes but the gametocyte specific proteins such as kinesin-8B, kinesin-15, kinesin-13, and PF16 are unique PIPs in *Plasmodium*[9,48]. These findings are consistent with our transcriptomic analysis of PP1PTD showing modulated expression of motor proteins in male gametocytes.

Analysis of PP1 during the zygote to ookinete transformation showed an additional location at the nascent apical end during the early stages of ookinete differentiation, which may help define

the cell's polarity. This suggestion is supported by the consequence of PP1PTD gene knock down in which numerous underdeveloped ookinetes with a long, thin protrusion attached to the main cell body were observed. This idea is also substantiated by a recent study showing that apical-basal polarity in *Drosophila* is controlled by PP1-mediated, and SDS22-dependent, dephosphorylation of LGL, an actomyosin-associated protein[62]. However, it is important to note that the suggested role in cell polarity is based on the observation that after fertilisation only a few abnormal ookinetes are formed, which have the morphologically distinct elongated apical end. Our transcriptomic analysis of PP1PTD-gametocytes showed modulation of genes for several organelle markers such as CTRP and SOAP[63,64], polarity markers such as SAS6L and IMC proteins[65,66], as well as genes involved in gliding motility including other IMC proteins, MyoA, GAPs, GAPMs and some ookinete specific proteins. These proteins are important for maintenance of cell shape, gliding motility and mosquito gut wall invasion by ookinetes. These results suggest that PP1 may be involved in the phosphoregulation of proteins involved in defining polarity and maintenance of ookinete shape.

In conclusion, PP1 is a constitutively expressed phosphatase, distributed throughout the cell but enriched in the nucleus, and associated with the kinetochore during mitosis and meiosis, with a role in the regulation of mitosis and meiosis throughout the *Plasmodium* life-cycle.

## Methods

**Ethics statement.** The animal work performed in the UK passed an ethical review process and was approved by the United Kingdom Home Office. Work was carried out under UK Home Office Project Licenses (30/3248 and PDD2D5182) in accordance with the United Kingdom 'Animals (Scientific Procedures) Act 1986'. Six- to eight-week-old female CD1 outbred mice from Charles River laboratories were used for all experiments in the UK.

**Generation of transgenic parasites.** GFP-tagging vectors were designed using the p277 plasmid vector and transfected as described previously[23]. A schematic representation of the endogenous *pp1* locus (PBANKA_1028300), the constructs and the recombined *pp1* locus can be found in Fig S1a. For GFP-tagging of PP1 by single crossover homologous recombination, a region of *pp1* downstream of the ATG start codon was used to generate the construct. For the genotypic analyses, a diagnostic PCR reaction was performed as outlined in Fig. S1a. Primer 1 (intP6tg) and primer 2 (ol492) were used to determine correct integration of the *gfp* sequence at the targeted locus. For western blotting, purified gametocytes were lysed using lysis buffer (10 mM TrisHCl pH 7.5, 150 mM NaCl, 0.5 mM EDTA and 1% NP-40). The lysed samples were boiled for 10 min at 95 °C after adding Laemmli buffer and were centrifuged at maximum speed (13,000×$g$) for 5 min. The samples were electrophoresed on a 4–12% SDS-polyacrylamide gel. Subsequently, resolved proteins were transferred to nitrocellulose membrane (Amersham Biosciences). Immunoblotting was performed using the Western Breeze Chemiluminescence Anti-Rabbit kit (Invitrogen) and anti-GFP polyclonal antibody (Invitrogen) at a dilution of 1:1250, according to the manufacturer's instructions.

To study the function of PP1, we used two conditional knock down systems; a promoter exchange/trap using double homologous recombination (PP1PTD) and an auxin inducible degron (PP1AID) system. The PP1AID construct was derived from the p277 plasmid, where the GFP sequence was excised following digestion with AgeI and NotI restriction enzymes and replaced with an AID/HA coding sequence. The AID-HA sequence was PCR amplified (using primers: 5′-CCCCAGACGTCGGATCCAATGATGGGCAGTGTCGAGCT-3′ and 5′-ATATAAGTAAGAAAAACGGCTTAAGCGTAATCTGGA-3′) from the GW-AID/HA plasmid (http://plasmogem.sanger.ac.uk/). Fragments were assembled following the Gibson assembly protocol to generate the PP1-AID/HA transfection plasmid that was transfected in the 615 line. Conditional degradation of PP1-AID/HA was performed as described previously[67]. A schematic representation of the endogenous *pp1* locus (PBANKA_1028300), the constructs and the recombined *pp1* locus can be found in Fig S2a. A diagnostic PCR was performed for *pp1* gene knockdown parasites as outlined in Fig. S2a. Primer 1 and Primer 3 were used to determine successful integration of the targeting construct at the 3′ gene locus (Fig S2b). Primer 1 and Primer 2 were used as controls (Fig S2b).

The conditional knockdown construct PP1-PTD was derived from $P_{ama1}$ (pSS368) where *pp1* was placed under the control of the *ama1* promoter, as described previously[68]. A schematic representation of the endogenous *pp1* locus, the constructs and the recombined *pp1* locus can be found in Fig S2d. A diagnostic PCR was performed for *pp1* gene knockdown parasites as outlined in Fig. S2d. Primer 1 (5′-intPTD36) and Primer 2 (5′-intPTD) were used to determine

successful integration of the targeting construct at the 5′ gene locus. Primer 3 (3′-intPTD) and Primer 4 (3′-intPTama1) were used to determine successful integration for the 3′ end of the gene locus (Fig. S2e). All the primer sequences can be found in Supplementary Data 4. *P. berghei* ANKA line 2.34 (for GFP-tagging) or ANKA line 507cl1 expressing GFP (for the knockdown construct) parasites were transfected by electroporation[47].

**Purification of schizonts and gametocytes.** Blood cells obtained from infected mice (day 4 post infection) were cultured for 11 h and 24 h at 37 °C (with rotation at 100 rpm) and schizonts were purified the following day on a 60% v/v NycoDenz (in PBS) gradient, (NycoDenz stock solution: 27.6% w/v NycoDenz in 5 mM Tris-HCl, pH 7.20, 3 mM KCl, 0.3 mM EDTA).

The purification of gametocytes was achieved by injecting parasites into phenylhydrazine treated mice[69] and enriched by sulfadiazine treatment after 2 days of infection. The blood was collected on day 4 after infection and gametocyte-infected cells were purified on a 48% v/v NycoDenz (in PBS) gradient (NycoDenz stock solution: 27.6% w/v NycoDenz in 5 mM Tris-HCl, pH 7.20, 3 mM KCl, 0.3 mM EDTA). The gametocytes were harvested from the interface and activated.

**Live-cell imaging.** To examine PP1-GFP expression during erythrocyte stages, parasites growing in schizont culture medium were used for imaging at different stages (ring, trophozoite, schizont and merozoite) of development. Purified gametocytes were examined for GFP expression and localisation at different time points (0, 1–15 min) after activation in ookinete medium[43]. Zygote and ookinete stages were analysed throughout 24 h of culture. Images were captured using a ×63 oil immersion objective on a Zeiss Axio Imager M2 microscope fitted with an AxioCam ICc1 digital camera (Carl Zeiss, Inc).

**Generation of dual tagged parasite lines.** The PP1-GFP parasites were mixed with NDC80-cherry and MyoA-cherry parasites in equal numbers and injected into mice. Mosquitoes were fed on mice 4–5 days after infection when gametocyte parasitaemia was high. These mosquitoes were checked for oocyst development and sporozoite formation at days 14 and 21 after feeding. Infected mosquitoes were then allowed to feed on naïve mice and after 4–5 days, and the mice were examined for blood stage parasitaemia by microscopy with Giemsa-stained blood smears. In this way, some parasites expressed both PP1-GFP and NDC80-cherry; and PP1-GFP and MyoA-cherry in the resultant gametocytes, and these were purified and fluorescence microscopy images were collected as described above.

**Parasite phenotype analyses.** Blood containing ~50,000 parasites of the PP1PTD line was injected intraperitoneally (i.p.) into mice to initiate infections. Asexual stages and gametocyte production were monitored by microscopy on Giemsa-stained thin smears. Four to five days post infection, exflagellation and ookinete conversion were examined with a Zeiss AxioImager M2 microscope (Carl Zeiss, Inc) fitted with an AxioCam ICc1 digital camera[70]. To analyse mosquito transmission, 30–50 *Anopheles stephensi* SD 500 mosquitoes were allowed to feed for 20 min on anaesthetised, infected mice with an asexual parasitaemia of 15% and a comparable number of gametocytes as determined on Giemsa-stained blood films. To assess mid-gut infection, ~15 guts were dissected from mosquitoes on day 14 post feeding, and oocysts were counted on an AxioCam ICc1 digital camera fitted to a Zeiss AxioImager M2 microscope using a ×63 oil immersion objective. On day 21 post-feeding, another 20 mosquitoes were dissected, and their guts crushed in a loosely fitting homogenizer to release sporozoites, which were then quantified using a haemocytometer or used for imaging. Mosquito bite back experiments were performed 21-days post-feeding using naive mice, and blood smears were examined after 3–4 days.

**Electron microscopy.** Gametocytes activated for 6 min and 30 min were fixed in 4% glutaraldehyde in 0.1 M phosphate buffer and processed for electron microscopy[71]. Briefly, samples were post fixed in osmium tetroxide, treated en bloc with uranyl acetate, dehydrated and embedded in Spurr's epoxy resin. Thin sections were stained with uranyl acetate and lead citrate prior to examination in a JEOL JEM-1400 electron microscope (JEOL Ltd, UK)

**Quantitative real-time PCR analyses.** RNA was isolated from gametocytes using an RNA purification kit (Stratagene). cDNA was synthesised using an RNA-to-cDNA kit (Applied Biosystems). Gene expression was quantified from 80 ng of total RNA using a SYBR green fast master mix kit (Applied Biosystems). All the primers were designed using the primer3 software (https://primer3.ut.ee/). Analysis was conducted using an Applied Biosystems 7500 fast machine with the following cycling conditions: 95 °C for 20 s followed by 40 cycles of 95 °C for 3 s; 60 °C for 30 s. Three technical replicates and three biological replicates were performed for each assayed gene. The *hsp70* (PBANKA_081890) and *arginyl-t RNA synthetase* (PBANKA_143420) genes were used as endogenous control reference genes. The primers used for qPCR can be found in Supplementary Data 3.

**Transcriptome study using RNA-seq.** For RNA extraction, parasite samples were passed through a plasmodipur column to remove host DNA contamination prior

to RNA isolation. Total RNA was extracted from activated gametocytes and schizonts of WT-GFP and PP1PTD parasites (two biological replicates each) using an RNeasy purification kit (Qiagen). RNA was vacuum concentrated (SpeedVac) and transported using RNA-stable tubes (Biomatrica). Strand-specific 354mRNA sequencing was performed on total RNA and using TruSeq stranded mRNA sample prep 355kit LT (Illumina)[72]. Libraries were sequenced using an Illumina Hiseq 4000 sequencing platform with paired-end 150 bp read chemistry. The quality of the raw reads was assessed using FASTQC (http://www.bioinformatics.babraham.ac.uk/projects/fastqc). Low-quality reads and Illumina adaptor sequences from the read ends were removed using Trimmomatic R[73]. Processed reads were mapped to the *P. beghei ANKA* reference genome (release 40 in PlasmoDB - http://www.plasmoddb.org) using Hisat2[74] (V 2.1.0) with parameter "—rna-strandness FR". Counts per feature were estimated using FeatureCounts[75]. Raw read counts data were converted to counts per million (cpm) and genes were excluded if they failed to achieve a cpm value of 1 in at least one of the three replicates performed. Library sizes were scale-normalised by the TMM method using EdgeR software[76] and further subjected to linear model analysis using the voom function in the limma package[77]. Differential expression analysis was performed using DeSeq2[78]. Genes with a fold-change greater than two and a false discovery rate corrected *p*-value (Benjamini–Hochberg procedure) < 0.05 were considered to be differentially expressed. Functional groups shown in Fig. 6c were inferred from annotations available in PlasmoDB: Release 49 (https://plasmodb.org/plasmo/app).

**Immunoprecipitation and mass spectrometry**. Schizonts, following 11 h and 24 h, respectively in in vitro culture, and male gametocytes 11 min post activation were used to prepare cell lysates. Purified parasite pellets were crosslinked using formaldehyde (10 min incubation with 1% formaldehyde, followed by 5 min incubation in 0.125 M glycine solution and three washes with phosphate-buffered saline (PBS) (pH, 7.5). Immunoprecipitation was performed using crosslinked protein and a GFP-Trap®_A Kit (Chromotek) following the manufacturer's instructions. Proteins bound to the GFP-Trap®_A beads were digested using trypsin and the peptides were analysed by LC-MS/MS. Briefly, to prepare samples for LC-MS/MS, wash buffer was removed, and ammonium bicarbonate (ABC) was added to beads at room temperature. We added 10 mM TCEP (Tris-(2-carboxyethyl) phosphine hydrochloride) and 40 mM 2-chloroacetamide (CAA) and incubation was performed for 5 min at 70 °C. Samples were digested using 1 μg Trypsin per 100 μg protein at room temperature overnight followed by 1% TFA addition to bring the pH into the range of 3–4 before mass spectrometry.

**Statistics and reproducibility**. All statistical analyses were performed using GraphPad Prism 8 (GraphPad Software). An unpaired *t*-test and two-way anova test were used to examine significant differences between wild-type and mutant strains for qRT-PCR and phenotypic analysis accordingly. For all experiments, biological replicates were defined as data generated from parasites injected into at least three mice.

**Reporting summary**. Further information on research design is available in the Nature Research Reporting Summary linked to this article.

## Data availability

RNA Sequence reads have been deposited in the NCBI Sequence gene expression omnibus with the accession number GSE164175. The mass spectrometry proteomics data have been deposited to the ProteomeXchange Consortium with the dataset identifier PXD023571 and 10.6019/PXD023571.

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

## Acknowledgements

We thank Julie Rodgers for helping to maintain the insectary and other technical works, and the personnel at the Bioscience Core Laboratory (BCL) in KAUST for sequencing the RNA samples and producing the raw datasets. This work was supported by: MRC UK (G0900278, MR/K011782/1, MR/N023048/1) and BBSRC (BB/N017609/1) to R.T. and M.Z.; the Francis Crick Institute (FC001097), the Cancer Research UK (FC001097), the UK Medical Research Council (FC001097), and the Wellcome Trust (FC001097) to A.A.H.; the Swiss National Science Foundation project grant 31003A_179321 to M.Br; a faculty baseline fund (BAS/1/1020-01-01) and a Competitive Research Grant (CRG) award from OSR (OSR-2018-CRG6-3392) from the King Abdullah University of Science and Technology to A.P.; M.Br is an INSERM and EMBO young investigator. This research was funded in whole, or in part, by the Wellcome Trust [FC001097]. For the purpose of Open Access, the author has applied a CC BY public copyright licence to any Author Accepted Manuscript version arising from this submission.

## Author contributions

R.T. and M.Z. conceived and designed all experiments. R.T., M.Z., R.P., D.B., D.S.G., and G.K. performed the GFP tagging and conditional knockdown with promoter trap experiments. R.R. and M.Br generated and characterised the PP1-AID/HA line. M.Z., R.P., G.K., D.B., and R.T. performed protein pull-down experiments. A.R.B. performed mass spectrometry. A.S., R.N. and A.P. performed RNA sequencing (RNA-seq). D.J.P.F. and S.V. performed electron microscopy. R.T., A.A.H., M.Z., D.S.G., A.S., R.N., A.P. and D.J.P.F. analysed the data. M.Z., D.S.G. and R.T. wrote the original draft. R.T., A.A.H., A.P., D.J.P.F. and M.B. edited and reviewed the manuscript and all other contributed to it.

## Competing interests

The authors declare no competing interests.
