## [Transparent Peer Review File · Communications Biology]

Reviewers' comments:

Reviewer #1 (Remarks to the Author):

The manuscript by Zeeshan M et al presented a study of protein phosphatase 1 involvement in parasite sexual stages cell division (mitosis and meiosis) in murine malaria model (*P. berghei*). This is a well-performed study with impressive multichannel fluorescence images in different sexual stages, which are difficult to work. An important effort was made using additional strategies for functional characterization of phosphatase, such as conditional gene knockdown, electron microscopy, RNA-seq and immunoprecipitation. As a continuation of previous studies recently published by the authors (Zeeshan M. et al *J. Cell Sci* 2020; Pandey R et al 2020 *Cell Rep*) using similar strategies, the data contribution is relevant for understanding the signaling elements related to PP1 (i.e., cell distribution and protein partners), present in chromosome segregation during parasite division and sexual stages development.

There are some concerns about the manuscript listed below:

- Introduction or discussion section – the authors can include a mention about the potential antimalarial drug discovery for PfPP1 (the homology differences with mammals' phosphatases) and if some inhibitor is already studied.
- The relevant reference is missing (Lenne A. et al 2018 *Frontiers Microbiol.*).
- Why the authors did not try to perform interactomes of PP1 in others stages besides schizonts and gametocytes? The role of PP1 could be more complete.
- If possible it's interestingly the a usage of exogenous phosphatase inhibitors as pharmacological controls of knockdown parasites and chromosome segregation assays.
- There is a large amount of diffused fluorescence of PP1-GFP in cytosol besides kinetochore, could be an overexpression without intrinsic targeting information, improperly folded molecules, etc of PP1-GFP construct? (it is quite similar intensity in cytosol of all cell stages analyzed - sexual and asexual) Is necessary a better discussion about it, there is a cellular relevance or artifact? What assays the authors would perform for clarify this point?
- Lines 384-387: The reference is missing [23] Guterry DS. et al 2014 *Cell Host Microbe*.

Reviewer #2 (Remarks to the Author):

The authors investigate the protease PP1 in the rodent malaria parasite *Plasmodium berghei* and find that it is important for the correct and efficient formation of gametes and ookinetes, which are essential for infecting the mosquito. Prior work by others has shown that PP1 is important in the blood stages of the parasite, where genetic alterations are performed. Hence the authors could not simply delete the gene for PP1 but explored two different conditional methods, of which one worked. Their fluorescence and electron microscopy are of highest standards and the results of the paper are clearly delineated. Maybe the authors could add the quantification of the different structures observed at 30 min pa in wt vs PP1 knockdown to the figure. It might help the unfamiliar reader to immediately grasp the differences that can be observed in the figures. Maybe also a different alignment would help in this figure, with WT being on the left column and the PP1 knockdown on the right, 6 min the top row and 30 min the bottom row. But this might be also just personal choice.

Reviewer #3 (Remarks to the Author):

Tewari and colleagues report on a Plasmodium serine/threonine phosphatase to be very important for the progression of the sexual stages of the parasite, by affecting parasite growth and development within the mosquito vector. It affects male gametogenesis (few flagellate forms), and of the gametes that survive, fewer form ookinetes (which are mostly round and not banana-shaped), and no oocysts or sporozoites.

This is certainly an important and well-written report with a good experimental design and novel data.

I do not have any major problem with the manuscript or the data presented. I only have one concern and that is summarized in the title. While the data clearly show that this phosphatase is critical, it does not show which functions are affected prior to the observed phenotypes. Is it the formation of the mitotic spindle? mitosis output? chromosome condensation? Or any other hypotheses.

Moreover, the putative role in cell polarity is based on the fact that after fertilization few ookinetes are banana-shaped. Instead they are round. But the authors do not look for the location of organelles, or polarity markers. Is it possible that the format they take may be just a consequence of an abortive development.

As for a putative role in the oocyst mitosis within the mosquito - in the context of a parasite where the protein expression is diminished significantly before in development and has already given rise to some phenotypes such as less and deformed ookinetes, the observed defects in these mitoses may be only a consequence and not a direct impact at that stage.

So, I believe this is a good report on the impact of the lack of a phosphatase in Plasmodium life cycle but I would ask the authors to be more careful in their conclusions, that can certainly be discussed as possibilities, but should not be added as definitive conclusions in the title or abstract.

Response to reviewers' comments

We thank all the reviewers for their positive, encouraging and constructive comments. We address their comments and suggestions, point-by-point and in detail below. Each reviewer's comments are in normal font and are followed by our response highlighted in red italic font.

Reviewer #1:

The manuscript by Zeeshan M et al presented a study of protein phosphatase 1 involvement in parasite sexual stages cell division (mitosis and meiosis) in murine malaria model (*P. berghei*). This is a well-performed study with impressive multichannel fluorescence images in different sexual stages, which are difficult to work. An important effort was made using additional strategies for functional characterization of phosphatase, such as conditional gene knockdown, electron microscopy, RNA-seq and immunoprecipitation. As a continuation of previous studies recently published by the authors (Zeeshan M. et al *J. Cell Sci* 2020; Pandey R et al 2020 *Cell Rep*) using similar strategies, the data contribution is relevant for understanding the signaling elements related to PP1 (i.e., cell distribution and protein partners), present in chromosome segregation during parasite division and sexual stages development.

We thank the reviewer for these positive and encouraging comments

There are some concerns about the manuscript listed below:

- Introduction or discussion section – the authors can include a mention about the potential antimalarial drug discovery for PfPP1 (the homology differences with mammals' phosphatases) and if some inhibitor is already studied.

We have added additional text on this topic in the Introduction (lines 81-85, lines 101-104) and Discussion (line 372).

- The relevant reference is missing (Lenne A. et al 2018 *Frontiers Microbiol.*).

This reference has been added on line 101.

- Why the authors did not try to perform interactomes of PP1 in others stages besides schizonts and gametocytes? The role of PP1 could be more complete.

This is a valid point and the additional work would have offered a more complete picture if it had been feasible and possible to perform it; however, since our focus was on mitotic progression during schizogony and gametogony, we felt it necessary to obtain only the interactomes present during these stages, with further analyses of other life-cycle stages beyond the scope of this study. Furthermore, the additional experimental work to look at other stages would not change the message of the manuscript. While both schizogony and gametogony stages offer ample material on which to reproducibly perform these analyses, this is not the case for all stages of the life cycle. In the present Covid situation it would be very difficult to perform and standardise such experiments using the mouse unit, the insectary and the mass spectrometry facility.

- If possible it's interestingly the usage of exogenous phosphatase inhibitors as pharmacological controls of knockdown parasites and chromosome segregation assays.

We agree with the reviewer that it would be interesting to determine whether of PP1 inhibitors such as PP1-12 cause defects in chromosome segregation and that, in principle, specific inhibitors could be used to complement the PP1 knockdown assays. However, we consider that experimental validation of on-target specificity of any potential PP1 inhibitor would be both necessary and very laborious, and that such a drug discovery and validation study would be beyond the scope of this work. We have used the PTD and AID systems in other studies (Balestra et al Elife 2020; Pandey et al 2020 Cell Rep), and believe that these approaches represent the optimal way to investigate PP1 function, using knockdown experiments since the PP1 gene is refractory to deletion (Guttery D. et al. 2014 Cell Host Microbe).

- There is a large amount of diffused fluorescence of PP1-GFP in cytosol besides kinetochore, could be an overexpression without intrinsic targeting information, improperly folded molecules, etc of PP1-GFP construct? (it is quite similar intensity in cytosol of all cell stages analyzed - sexual and asexual) Is necessary a better discussion about it, there is a cellular relevance or artifact? What assays the authors would perform for clarify this point?

In our previous phosphatome study, GFP-fluorescence was observed in the cytosol of asexual blood stages and non-activated gametocytes (Guttery et al. 2014. Cell host Microbe). It was also observed in this study prior to gametocyte activation. Studies from other authors have also described PP1 in the cytosol (Daher, W. et al. 2006. Mol Micro and Paul et al. 2020. Nat Commun). We suggest in the Discussion (lines 399-403) that upon activation of male gametocytes and during mitotic progression, PP1-GFP is recruited to the kinetochores in order to perform a specific function, but residual PP1-GFP also remains in the cytosol since PP1 has many substrates and hence only subset of them may be modified in these stages.

- Lines 384-387: The reference is missing [23] Guttery DS. et al 2014 Cell Host Microbe.

We thank the reviewer for highlighting this omission, and have added the reference accordingly.

Reviewer #2:

The authors investigate the protease PP1 in the rodent malaria parasite Plasmodium berghei and find that it is important for the correct and efficient formation of gametes and ookinetes, which are essential for infecting the mosquito. Prior work by others has shown that PP1 is important in the blood stages of the parasite, where genetic alterations are performed. Hence the authors could not simply delete the gene for PP1 but explored two different conditional methods, of which one worked. Their fluorescence and electron microscopy are of highest standards and the results of the paper are clearly delineated. Maybe the authors could add the quantification of the different structures observed at 30 min pa in wt vs PP1 knockdown to the figure. It might help the unfamiliar reader to immediately grasp the differences that can be observed in the figures. Maybe also a different alignment would help in this figure, with WT being on the left column and the PP1 knockdown on the right, 6 min the top row and 30 min the bottom row. But this might be also just personal choice.

We thank the reviewer for their positive comments. As suggested, we have added quantitative data for early and late stages of gametogony at 30 mins post activation, for both WT and PP1-PTD parasites. We have also modified the figure according to the reviewer's suggestion.

Reviewer #3:

Tewari and colleagues report on a Plasmodium serine/threonine phosphatase to be very important for the progression of the sexual stages of the parasite, by affecting parasite growth and development within the mosquito vector. It affects male gametogenesis (few flagellate forms), and of the gametes that survive, fewer form ookinetes (which are mostly round and not banana-shaped), and no oocysts or sporozoites.

This is certainly an important and well-written report with a good experimental design and novel data.

We thank the reviewer for these positive comments.

I do not have any major problem with the manuscript or the data presented. I only have one concern and that is summarized in the title. While the data clearly show that this phosphatase is critical, it does not show which functions are affected prior to the observed phenotypes. Is it the formation of the mitotic spindle? mitosis output? chromosome condensation? Or any other hypotheses.

We thank the reviewer for highlighting this point and have changed the title to “Protein Phosphatase 1 regulates atypical mitotic and meiotic division in Plasmodium” to reflect this concern. We think that PP1 is important both for mitotic spindle function and chromosome condensation as observed in the ultrastructure studies.

Moreover, the putative role in cell polarity is based on the fact that after fertilization few ookinetes are banana-shaped. Instead they are round. But the authors do not look for the location of organelles, or polarity markers. Is it possible that the format they take may be just a consequence of an abortive development.

This is a valid point. We have now added further detail about this in the discussion (lines 445-449). Since the gene knockdown already affects male gamete formation then the effect on fertilisation and zygote formation is reduced. The few zygotes that do differentiate into abnormal ookinetes may be affected due to the mis-regulation of a number of polarity and motility genes, including those coding for IMC proteins, myosin and GAP50.

As for a putative role in the oocyst mitosis within the mosquito - in the context of a parasite where the protein expression is diminished significantly before in development and has already given rise to some phenotypes such as less and deformed ookinetes, the observed defects in these mitoses may be only a consequence and not a direct impact at that stage.

We thank the reviewer for this comment, however we do not see oocyst development in the knockdown mutant. We agree that since there are very few abnormal ookinetes produced, the primary effects of the gene knockdown are most likely at an earlier stage.

So, I believe this is a good report on the impact of the lack of a phosphatase in Plasmodium life cycle but I would ask the authors to be more careful in their conclusions, that can certainly be discussed as possibilities, but should not be added as definitive conclusions in the title or abstract.

We thank the reviewer for their constructive comments and have modified the Abstract and conclusions accordingly.

REVIEWERS' COMMENTS:

Reviewer #1 (Remarks to the Author):

The questions were answered accordingly and improving the text. The pandemic situation hindered any other complementary trials.

Reviewer #2 (Remarks to the Author):

Thanks for adding the additional quantification. Lovely paper.

Reviewer #3 (Remarks to the Author):

The authors have fully responded to all my comments and in my opinion the manuscript deserves to be published.